# BORN AGAIN NEURAL RANKERS

## ABSTRACT

We introduce Born Again neural Rankers (BAR) in the Learning to Rank (LTR) setting, where student rankers, trained in the Knowledge Distillation (KD) framework, are parameterized identically to their teachers. Unlike the existing ranking distillation work which pursues a good trade-off between performance and efficiency, BAR adapts the idea of Born Again Networks (BAN) to ranking problems and significantly improves ranking performance of students over the teacher rankers without increasing model capacity. The key differences between BAR and common distillation techniques for classification are: (1) an appropriate teacher score transformation function, and (2) a novel listwise distillation framework. Both techniques are specifically designed for ranking problems and are rarely studied in the knowledge distillation literature. Using the state-of-the-art neural ranking structure, BAR is able to push the limits of neural rankers above a recent rigorous benchmark study and significantly outperforms traditionally strong gradient boosted decision tree based models on 7 out of 9 key metrics, the first time in the literature. In addition to the strong empirical results, we give theoretical explanations on why listwise distillation is effective for neural rankers.

## 1    INTRODUCTION

Learning to rank (LTR) has become an essential component for many real-world applications such as search (Liu, 2009). In ranking problems, the focus is on predicting the relative order of a list of items given a query. It is thus different from classification problems whose goal is to predict the class label of a single item. While many neural machine learning techniques have been proposed recently, they are mostly for classification problems. Given the difference between ranking and classification problems, it is interesting to study how neural techniques can be used for ranking problems.

Knowledge distillation (KD) (Hinton et al., 2015; Gou et al., 2020) is one of such recently popular techniques. The initial goal of KD is to pursue a good trade-off between performance and efficiency. Given a high-capacity teacher model with desired high performance, a more compact student model is trained using teacher labels (Heo et al., 2019; Sun et al., 2019; Sanh et al., 2019). Student models trained with KD usually work better than those trained from original labels without teacher guidance. How to effectively apply KD to ranking problems is not straightforward and has not been well-studied yet, for the following reasons:

First, teacher models in classification typically predict a probability distribution over all classes. Such "dark knowledge" is believed to be a key reason why KD works (Hinton et al., 2015). However, a teacher model in ranking does not convey such distribution over pre-defined classes, as ranking models only care about the relative orders of items and simply output a single score for each item in a possibly unbounded candidate list. Ranking scores are typically neither calibrated (comparing to the probabilistic interpretation of classification scores (Guo et al., 2017)) nor normalized (comparing to summing up to one for classification scores over all classes for most popular losses (Hinton et al., 2015)). Thus directly using the outputs of a teacher model as labels for ranking tasks with existing KD methods may be less optimal.

Second, listwise information over all input items should be considered to achieve the best ranking performance, since the goal is to infer the relative orders among them. For example, listwise losses have been shown to be more effective than other alternatives for LTR problems (Cao et al., 2007). On the other hand, classification tasks almost universally treat each item independently based on the

i.i.d. assumption. It is interesting to also consider listwise frameworks when studying KD for ranking problems, which has not been explored in the literature, to the best of our knowledge.

Third, though there is a consensus in the classification KD literature that given no severe over-fitting, larger teacher models usually work better (He et al., 2016), recent works show that for traditional LTR problems, it is hard for larger models to achieve better performance (Bruch et al., 2019; Qin et al., 2021) due to the difficulty of applying standard techniques such as data augmentation (compared to image rotation in computer vision (Xie et al., 2020)) and the lack of very large human-labeled ranking datasets. This makes it harder to use the standard KD setting to improve the performance of ranking models by distilling from a very large teacher model.

Thus, KD techniques need special adjustments for LTR problems. In this paper, inspired by the Born Again Networks (BAN) (Furlanello et al., 2018) in which student models are configured with the same capacity as teachers and are shown to outperform teachers for classification problems, we study the BAN techniques for ranking problems. To this end, we propose Born Again neural Rankers (BAR) that train the student models using listwise distillation and properly transformed teacher scores, which can achieve new state-of-the-art ranking performance for neural rankers. While existing ranking distillation works such as (Tang & Wang, 2018; Reddi et al., 2021) require a more powerful teacher model and focus on performance-efficiency trade-offs, the primary goal of our paper is to improve ranking performance over state-of-the-art teacher rankers.

In summary, our contributions are as follows:

- We propose Born Again neural Rankers (BAR) for learning to rank. This is the first knowledge distillation work that targets for *better* ranking performance without increasing the model capacity.
- We show that the key success factors of BAR are (1) an appropriate teacher score transformation function, and (2) a ranking specific listwise distillation loss. Both design choices are tailored for LTR problems and are rarely studied in the knowledge distillation literature.
- We provide new theoretical explanations on why BAR works better than other alternatives. This contributes to both the general knowledge distillation and ranking distillation research.
- We verify our hypothesis on rigorous public LTR benchmarks and show that BAR is able to significantly improve upon state-of-the-art neural teacher rankers.

## 2 RELATED WORK

Knowledge distillation has been a popular research topic recently in several areas such as image recognition (Hinton et al., 2015; Romero et al., 2014; Park et al., 2019), natural language understanding (Sanh et al., 2019; Jiao et al., 2019; Aguilar et al., 2020), and neural machine translation (Kim & Rush, 2016; Chen et al., 2017; Tan et al., 2019) as a way to generate compact models, achieving good performance-efficiency trade-offs. As we mentioned in the Introduction, both the classical setting (e.g., using pointwise losses on i.i.d. data) and theoretical analysis (e.g., "dark knowledge" among classes) for classification tasks may not be optimal for the ranking setting.

The major goal of this work is to push the limits of neural rankers on rigorous benchmarks. Since the introduction of RankNet (Burges et al., 2005) over a decade ago, only recently, neural rankers were shown to be competitive with well-tuned Gradient Boosted Decision Trees (GBDT) on traditional LTR datasets (Qin et al., 2021). We build upon (Qin et al., 2021) and show that BAR can further push the state-of-the-art of neural rankers on rigorous benchmarks. We also provide extra experiments to show listwise distillation helps neural ranking in other settings.

We are motivated by Born Again Networks (BAN) that were introduced by (Furlanello et al., 2018). BAN is the first work that shows better performance can be achieved by parameterizing the student model the same as the teacher. However, BAN only focuses on classification and direct application of BAN does not help LTR problems. Building upon the general "born again" ideas (Zhang et al., 2019; Clark et al., 2019), our contribution in this paper is in developing specific new techniques and theory that make these ideas applicable for the important LTR setting.

Another closely related work is Ranking Distillation (RD) (Tang & Wang, 2018), since it studies knowledge distillation for ranking. There are several marked differences between our work and RD. First, RD focuses on performance-efficiency trade-offs, where the student usually underperforms the

teacher in terms of ranking metrics (Reddi et al., 2021), while we focus on outperforming the teacher. Second, the work only uses pointwise logistic loss for distillation. The authors state that "We also tried to use a pair-wise distillation loss when learning from teacher's top-K ranking. However, the results were disappointing", without going into more detail or explore listwise approaches, which we show are the key success factor. Also, Tang and Wang (Tang & Wang, 2018) use a hyperparameter K: items ranked as top K are labeled as positive and others as negative for distillation. Besides only working for binary datasets, this setting is not very practical since real-world ranking lists may have very different list sizes or number of relevant items. Our method does not require such a parameter. Furthermore, they only evaluated their methods on some recommender system datasets where only user id, item id, and binary labels are available. It is not a typical LTR setting, so the effectiveness of RD over state-of-the-art neural rankers is unclear.

## 3 BACKGROUND ON LEARNING TO RANK

For LTR problems, the training data can be represented as a set $\Psi = \{(\mathbf{x}, \mathbf{y}) \in \chi^n \times \mathcal{R}^n)\}$, where $\mathbf{x}$ is a list of $n$ items $x_i \in \chi$ and $\mathbf{y}$ is a list of $n$ relevance labels $y_i \in \mathcal{R}$ for $1 \le i \le n$. We use $\chi$ as the universe of all items. In traditional LTR problems, each $x_i$ corresponds to a query-item pair and is represented as a feature vector in $\mathcal{R}^k$ where $k$ is the number of feature dimensions. With slight abuse of notation, we also use $x_i$ as the feature vector and say $\mathbf{x} \in \mathcal{R}^{n \times k}$. The objective is to learn a function that produces an ordering of items in $\mathbf{x}$ so that the utility of the ordered list is maximized, that is, the items are ordered by decreasing relevance.

Most LTR algorithms formulate the problem as learning a ranking function to score and sort the items in a list. As such, the goal of LTR boils down to finding a parameterized ranking function $f(\cdot; \theta) : \chi^n \to \mathcal{R}^n$, where $\theta$ denotes the set of trainable parameters, to minimize the empirical loss:

$$\mathcal{L}(f(\cdot; \theta)) = \frac{1}{|\Psi|} \sum_{(\mathbf{x}, \mathbf{y}) \in \Psi} l(\mathbf{y}, f(\mathbf{x}; \theta)), \tag{1}$$

where $l(\cdot)$ is the loss function on a single list.

There are many existing ranking metrics such as NDCG and MAP used in LTR problems. A common property of these metrics is that they are rank-dependent and place more emphasis on the top ranked items. For example, the widely used NDCG metric is defined as

$$NDCG(\pi_f, \mathbf{y}) = \frac{DCG(\pi_f, \mathbf{y})}{DCG(\pi^*, \mathbf{y})}, \tag{2}$$

where $\pi_f$ is a ranked list induced by the ranking function $f(\cdot; \theta)$ on $\mathbf{x}$, $\pi^*$ is the ideal list (where $\mathbf{x}$ is sorted by decreasing $\mathbf{y}$), and $DCG$ is defined as:

$$DCG(\pi, \mathbf{y}) = \sum_{i=1}^{n} \frac{2^{y_i} - 1}{\log_2(1 + \pi(i))}. \tag{3}$$

In practice, the truncated version that only considers the top-k ranked items, denoted as NDCG@k, is often used.

## 4 BORN AGAIN NEURAL RANKERS

### 4.1 THE GENERAL FORMULATION OF BAR

In addition to the original training data $\{(\mathbf{x}, \mathbf{y}) \in \chi^n \times \mathcal{R}^n)\}$, we assume there is a well-trained teacher model $f(\cdot; \theta^t)$. The goal of BAR is to train a student model $f(\cdot; \theta^s)$, where $\theta^s$ is parameterized the same as $\theta^t$, with the following loss:

$$\mathcal{L}(f(\cdot; \theta^s)) = \frac{1}{|\Psi|} \sum_{(\mathbf{x}, \mathbf{y}) \in \Psi} \left( (1 - \alpha) \times l(\mathbf{y}, f(\mathbf{x}; \theta^s)) + \alpha \times l_{\mathrm{B}}(g(f(\mathbf{x}; \theta^t)), f(\mathbf{x}; \theta^s)) \right), \tag{4}$$

where $\alpha \in [0, 1]$ is a weighting factor between the two losses, $l_{\mathrm{B}}$ is the additional Born Again loss, and $g(\cdot)$ is a transformation function for teacher scores. The practical necessity of this transform

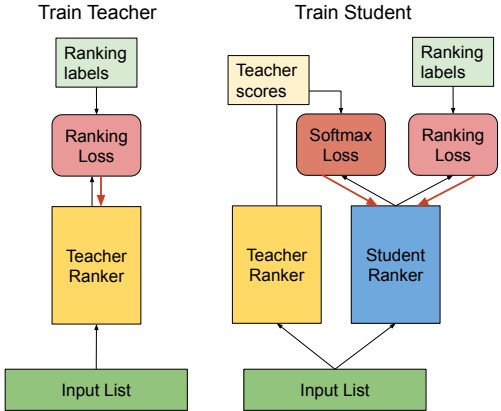

Figure 1: An illustration of the BAR framework. Black arrows show the feed-forward paths and red arrows direct the backward gradients. The teacher ranker can be trained with any ranking loss functions. The student model uses a multi-objective architecture that adds a listwise Softmax loss over teacher scores, in addition to the original ranking loss.

function is further analyzed in Section 4.3. The value of $\alpha$ can be tuned and we found that a wide range of $\alpha$ works well in our experiments.

An illustration of the BAR framework is shown in Fig. 1. The BAR framework uses a multi-objective setting for student score $f(\mathbf{x}; \theta^s)$. In addition to the original loss function, BAR adds an additional distillation loss between student and (transformed) teacher scores.

## 4.2 LOSS FUNCTIONS

Both the original loss function $l(\cdot, \cdot)$ and the born again loss $l_{\mathrm{B}}(\cdot, \cdot)$ can be any ranking loss, ranging from pointwise to pairwise and listwise losses (Liu, 2009). Generally, listwise and pairwise losses are more effective than pointwise ones (Cao et al., 2007). For example, the teacher models we leverage from (Qin et al., 2021) use the listwise Softmax cross entropy loss. Specifically, given the labels $\mathbf{y}$ and scores $\mathbf{s}$, the Softmax cross entropy loss is defined over a list of items:

$$l^{\mathrm{Softmax}}(\mathbf{y}, \mathbf{s}) = -\sum_{i=1}^{n} y_i \ln \frac{\exp(s_i)}{\sum_{j=1}^{n} \exp(s_j)} \tag{5}$$

where $n$ is the number of items in the ranking list.

In experiments we show that the original loss function can be more flexible (e.g., it can be a pointwise or listwise loss), but the born again loss is only effective when it is the listwise one.

## 4.3 TEACHER RANKER SCORE TRANSFORMATION

We formalize the necessity of properly transforming the scores from the teacher ranker. First, many popular ranking losses are translation invariant. For example, in the Softmax cross entropy loss, scores appear in the paired form of $s_i - s_j$:

$$l^{\mathrm{Softmax}}(\mathbf{y}, \mathbf{s}) = -\sum_{i=1}^{n} y_i \ln \frac{\exp(s_i)}{\sum_{j=1}^{n} \exp(s_j)} = -\sum_{i=1}^{n} y_i \ln \frac{1}{\sum_{j=1}^{n} \exp(s_j - s_i)}. \tag{6}$$

It is easy to see that the loss stays invariant after adding a constant to all scores. This is also the case for other popular ranking losses such as ApproxNDCG loss (Bruch et al., 2019) and the pairwise RankNet loss (Burges, 2010).

Second, since ranking scores are not normalized (compared to summing up to one for classification over all classes), teacher ranker's insensitivity to score scales may lead to numerical issues for distillation. For example, two documents can be ranked correctly with very close scores (e.g., 1.001 and 1.0) where the student may not differentiate them effectively, even if we subtract a constant from them. Thus, the scale of the teacher scores also need attention.

Third, unlike classification problems where predictions are non-negative probabilities, teacher ranker scores can be negative, which will make many ranking losses ill-behaved, such as any cross entropy based losses.

Putting these together, we propose a simple teacher score transformation function which parameterizes $g(\cdot)$ as an affine transformation function of teacher scores:

$$g(f(\mathbf{x}; \theta^t)) = \text{ReLU}(a \times f(\mathbf{x}; \theta^t) + b). \tag{7}$$

$\text{ReLU}(x) = \max(x, 0)$ is a safeguard to make sure the transformed teacher scores are non-negative. The slope $a(a > 0)$ and the intercept $b$ of the affine transformation are treated as hyperparameters. Note that we aim for a general formulation highlighting potential caveats of teacher ranking scores due to the difference from classification problem. The actual tuning depends on teacher ranker behavior and may be simple in practice. For example, when $a = 1$ and $b = 0$, $g(\cdot)$ becomes the standard ReLU function, which turns out to be effective for two out of three datasets used in our experiments.

### 4.4 ALTERNATIVES

We discuss possible alternatives that we will compare with in experiments. These alternatives are used to help highlight that BAR is referred to the specific setting where ranking loss with original labels, listwise distillation on teacher score, and tunable affine teacher score transformation are used. Our comparison is to help better understand what makes BAR effective in practice.

**Pointwise distillation.** Instead of the listwise loss, we can perform a pointwise loss for the distillation objective $l_\text{B}(\cdot, \cdot)$ in Eq 4:

$$l_\text{B}(g(f(\mathbf{x}; \theta^t)), f(\mathbf{x}; \theta^s)) = \sum_{i=1}^{n} l^p(g(f(x_i; \theta^t)), f(x_i; \theta^s)) \tag{8}$$

Note that a pointwise loss is decomposed across items independently, following the i.i.d. assumption for regression. Throughout the paper, we use the mean squared error loss as the pointwise loss:

$$l^\text{MSE}(g(f(\mathbf{x}; \theta^t)), f(\mathbf{x}; \theta^s))) = \sum_{i=1}^{n} \|g(f(x_i; \theta^t)) - f(x_i; \theta^s))\|^2, \tag{9}$$

since the original labels are graded with real values, and as we mentioned in Section 4.3, the range of teacher scores can be unbounded, even for binary labeled datasets.

**Teacher score only distillation.** Another way to distill teacher score is to ignore the original loss function using original labels ($\alpha = 1$ in Equation 4):

$$\mathcal{L}(f(\cdot; \theta^s)) = \frac{1}{|\Psi|} \sum_{(\mathbf{x}, \mathbf{y}) \in \Psi} l^\text{Softmax}(g(f(\mathbf{x}; \theta^t)), f(\mathbf{x}; \theta^s)). \tag{10}$$

The original BAN paper (Furlanello et al., 2018) shows that this single objective formulation works better than the two objective formulation in certain classification scenarios.

**Softmax teacher score transformation.** It is tempting to apply the Softmax transformation on teacher scores in each list, since it converts all labels to be non-negative while preserving their relative order. We study Softmax teacher score transform with temperatures, $\frac{\exp(f(x_i; \theta^t)/T)}{\sum_{j=1}^{n} \exp(f(x_j; \theta^t)/T)}$, in Section 5.6 and empirically show that it under-performs affine transformations.

## 5 EXPERIMENTS

We conduct experiments on three public LTR datasets to show that BAR achieves state-of-the-art performance for neural rankers. We also perform various ablation studies to better understand BAR.

### 5.1 EXPERIMENTAL SETUP

We follow the experimental setup of (Qin et al., 2021) that provides a rigorous LTR benchmark. Three widely adopted data sets for web search ranking are used: Web30K (Qin & Liu, 2013), Yahoo

(Chapelle & Chang, 2011), and Istella (Dato et al., 2016). The documents for each query were labeled with multilevel graded relevance judgments by human raters. We compare with a comprehensive list of benchmark methods.

$\lambda$MART$_{GBM}$ (Ke et al., 2017) and $\lambda$MART$_{RankLib}$ are the two GBDT-based implementations for LTR. RankSVM (Joachims, 2006) is a classic pairwise learning-to-rank model built on SVM. GSF (Ai et al., 2019) is a neural model using groupwise scoring function and fully connected layers. ApproxNDCG (Bruch et al., 2019) is a neural model with fully connected layers and a differentiable loss that approximates NDCG (Qin et al., 2010). DLCM (Ai et al., 2018) is an RNN based neural model that uses list context information to rerank a list of documents based on $\lambda$MART$_{RankLib}$ as in the original paper. SetRank (Pang et al., 2020) is a neural model using self-attention to encode the entire list and perform a joint scoring. SetRank$^{re}$ (Pang et al., 2020) is SetRank plus ordinal embeddings based on the initial document ranking generated by $\lambda$MART$_{RankLib}$ as in the original paper. DASALC (Qin et al., 2021) is the state-of-the-art neural rankers combining data transformation and augmentation, effective feature crosses, and listwise context. DASALC-ens is an ensemble of DASALC models, significantly outperforming the strong $\lambda$MART$_{GBM}$ baseline on 4 out of 9 major metrics.

For the main results of BAR, we obtain the teacher model scores and configurations of DASALC (not DASALC-ens) for each dataset from the authors. For the Web30K and Yahoo dataset, the teacher scores are used as they are (i.e., we use the identity function as the affine transformation in $g(\cdot)$). As the teacher scores for the Istella dataset are of larger range (mean = 28.4, std = 299.4) and are causing some numeric issues, we dampen them by using $g(f(\mathbf{x}; \theta^t)) = \text{ReLU}(\frac{1}{100} \cdot f(\mathbf{x}; \theta^t))$. Better results may be achieved by more tuning. The model architecture configurations, such as the number of hidden layers, are unchanged during BAR training. We simply use $\alpha = 0.5$ to assign equal weights between the original objective and distillation objective, and show its robustness in Section 5.5. We do perform hyper-parameter searches over dropout rate, learning rate, and data augmentation noise level on validation sets. Note that all BAR models are only born again once (instead of iteratively born again multiple times by treating previous student model as the new teacher), due to computation overhead and the lack of theoretical guarantee in the original BAN work. BAR is a single neural ranker. BAR-ens is an ensemble of $k$ rankers ($3 \leq k \leq 5$) that is tuned on each dataset with the same architecture from different runs. Note that the candidates in the ensemble for each data set are supervised by the same teacher.

## 5.2 RESEARCH QUESTIONS

We want to answer the following research questions by comparing with competitive methods on LTR benchmarks and performing ablation studies:

- RQ1: Is the BAR framework able to further push the limits of neural rankers over the state-of-the-art models?
- RQ2: Is the listwise distillation loss necessary for BAR to be effective for LTR problems, compared to the common pointwise distillation loss that follows the i.i.d. assumption?
- RQ3: Is the dual-objective architecture necessary for BAR, considering that (Furlanello et al., 2018) shows that using only the teacher objective works better in certain scenarios? How robust is the balancing parameter $\alpha$ between the two objectives?
- RQ4: Is the affine teacher score transformation more effective than the Softmax transformation?

## 5.3 MAIN RESULTS (RQ1)

Our main results are shown in Table 1. From this table, we can see that the models trained under the BAR framework can significantly push the limits over the state-of-the-art neural rankers without sacrificing efficiency. A single BAR model performs best among non-GBDT methods on 8 out of 9 metrics. BAR-ens universally outperforms DASALC-ens, and can significantly outperform $\lambda$MART$_{GBM}$ on 7 out of 9 metrics.

## 5.4 THE NECESSITY OF LISTWISE DISTILLATION (RQ2)

We compare with pointwise distillation in Table 2 on the Web30K dataset. We use the MSE loss and tune $\alpha$ in Eq. 4 for the datasets with graded relevance labels. Results on other datasets are consistent.

Table 1: Result on the Web30K, Yahoo, and Istella datasets. $^{\uparrow}$ means significantly better result, performed against $\lambda\text{MART}_{GBM}$ at the $p < 0.05$ level using a two-tailed $t$-test. Bold numbers are the highest in each column, italicized numbers are the highest in each column for single neural rankers.

| Models | Web30K NDCG@k | | | Yahoo NDCG@k | | | Istella NDCG@k | | |
|---|---|---|---|---|---|---|---|---|---|
| | @1 | @5 | @10 | @1 | @5 | @10 | @1 | @5 | @10 |
| $\lambda\text{MART}_{RankLib}$ | 45.35 | 44.59 | 46.46 | 68.52 | 70.27 | 74.58 | 65.71 | 61.18 | 65.91 |
| $\lambda\text{MART}_{GBM}$ | 50.73 | 49.66 | 51.48 | **71.88** | 74.21 | 78.02 | **74.92** | 71.24 | 76.07 |
| RankSVM | 30.10 | 33.50 | 36.50 | 63.70 | 67.40 | 72.60 | 52.69 | 50.41 | 55.29 |
| GSF | 41.29 | 41.51 | 43.74 | 64.29 | 68.38 | 73.16 | 62.24 | 59.68 | 65.08 |
| ApproxNDCG | 46.64 | 45.38 | 47.31 | 69.63 | 72.32 | 76.77 | 65.81 | 62.32 | 67.09 |
| DLCM | 46.30 | 45.00 | 46.90 | 67.70 | 69.90 | 74.30 | 65.58 | 61.94 | 66.80 |
| SetRank | 42.90 | 42.20 | 44.28 | 67.11 | 69.60 | 73.98 | 67.33 | 62.78 | 67.37 |
| SetRank$^{re}$ | 45.91 | 45.15 | 46.96 | 68.22 | 70.29 | 74.53 | 67.60 | 63.45 | 68.34 |
| DASALC | 50.95 | 50.92$^{\uparrow}$ | 52.88$^{\uparrow}$ | *70.98* | 73.76 | 77.66 | 72.77 | 70.06 | 75.30 |
| BAR | *51.71*$^{\uparrow}$ | *51.56*$^{\uparrow}$ | *53.57*$^{\uparrow}$ | 70.83 | *73.89* | *77.85* | *72.97* | *70.24* | *75.47* |
| ($\Delta$ over DASALC) | (+1.49%) | (+1.26%) | (+1.30%) | (-0.21%) | (+0.18%) | (+0.24%) | (+0.27%) | (+0.26%) | (+0.23%) |
| DASALC-ens | 51.89$^{\uparrow}$ | 51.72$^{\uparrow}$ | 53.73$^{\uparrow}$ | 71.24 | 74.07 | 77.97 | 74.40 | 71.32 | 76.44$^{\uparrow}$ |
| BAR-ens | **52.22**$^{\uparrow}$ | **52.12**$^{\uparrow}$ | **54.09**$^{\uparrow}$ | 71.45 | **74.61**$^{\uparrow}$ | **78.31**$^{\uparrow}$ | 74.51 | **71.51**$^{\uparrow}$ | **76.56**$^{\uparrow}$ |
| ($\Delta$ over DASALC-ens) | (+0.64%) | (+0.77%) | (+0.67%) | (+0.29%) | (+0.73%) | (+0.44%) | (+0.15%) | (+0.27%) | (+0.16%) |

| Method | | NDCG@k | | |
|---|---|---|---|---|
| Teacher | Student | @1 | @5 | @10 |
| Pointwise | / | 50.67 | 50.70 | 52.85 |
| | Pointwise | 50.33 | 50.47 | 52.60 |
| | Listwise | 50.88 | 51.16 | 53.26 |
| Listwise | / | 50.95 | 50.92 | 52.88 |
| | Pointwise | 49.78 | 49.89 | 52.02 |
| | Listwise | **51.71** | **51.56** | **53.57** |

Table 2: Results on the Web30K dataset comparing listwise vs pointwise teachers and distillation. "/" means directly using the teacher model (DASALC (Qin et al., 2021)) without distillation.

In this table, listwise teacher without distillation is the DASALC model, listwise teacher plus listwise student is the BAR model. We have the following observations:

- By comparing pointwise teacher with listwise teacher (DASALC), we confirm that listwise loss is indeed effective, regardless of model distillations.
- We can also see that when listwise distillation is used, it always outperforms the corresponding teacher model, even if the teacher model is a pointwise one. On the other hand, when pointwise distillation is applied, performance gets worse. This confirms the necessity of the listwise distillation loss.
- Pointwise distillation does not help even if the teacher model uses pointwise loss, eliminating the potential concern that it does not work due to inconsistency between the two objectives.

## 5.5 THE NECESSITY OF TWO OBJECTIVES (RQ3)

We study the balancing factor $\alpha$ in Eq. 4, with the special case ($\alpha = 1$) that only uses the teacher-only objective in Section 4.4. The results are shown in Table 3 and Figure 2. We can see that student models trained with BAR can outperform the teacher model ($\alpha = 0$) with a *wide* range of $\alpha$, as long as both objectives are used. The performance decreases significantly if the born again ranker is trained with the teacher objective alone ($\alpha = 1$), which is different from the observations for some classification problems in (Furlanello et al., 2018).

## 5.6 SOFTMAX SCORE TRANSFORMATION (RQ4)

We empirically show that Softmax score transformation under-performs the simple affine transformation in Figure 3 (Left) with varying the temperature $T$. A possible reason is that Softmax scores are normalized within each query and this may limit their power as labels for ranking problems. A better understanding of why certain teacher score transformations are more effective for ranking distillation is a promising future research direction.

| Method | Web30K NDCG@k | | |
|---|---|---|---|
| | @1 | @5 | @10 |
| Teacher model | 50.95 | 50.92 | 52.88 |
| BAR | **51.71** | **51.56** | **53.57** |
| Teacher-only obj | 50.45 | 50.53 | 52.54 |

Table 3: Results on the Web30K dataset comparing teacher model ($\alpha = 0$), BAR ($\alpha = 0.5$), and teacher-only objective born again rankers ($\alpha = 1$).

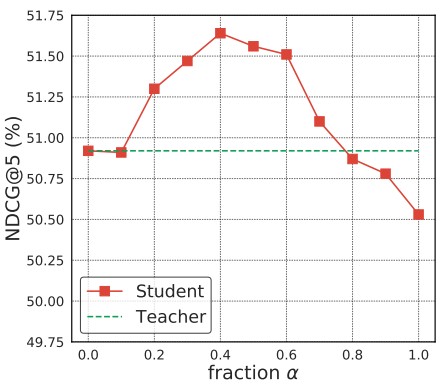

Figure 2: NDCG@5 with varying $\alpha$.

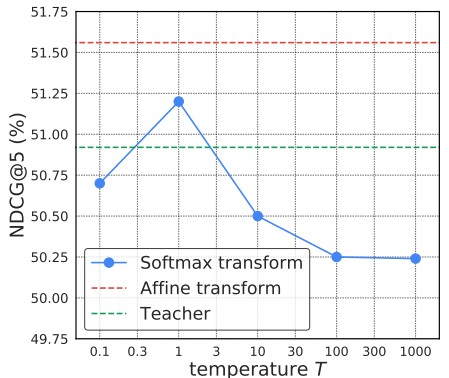

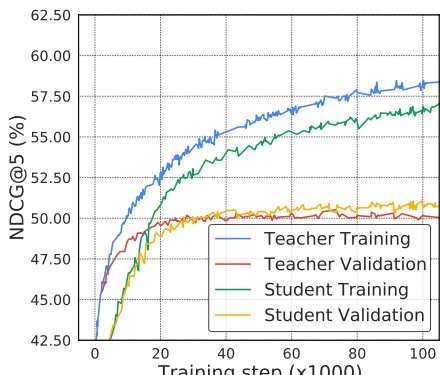

Figure 3: (Left) NDCG@5 on Web30K using Softmax score transformation with varying temperatures. (Right) An illustration of ranking metrics over training data and validation data for BAR (student) vs DASALC (teacher).

## 6 UNDERSTANDING BAR

Existing theories of Born Again Networks or knowledge distillation in general include (1) reweighting the data points by confidence (Furlanello et al., 2018), (2) incorporating dark knowledge from negative classes (Hinton et al., 2015), and (3) ensembling of "multi-view" features in student (Allen-Zhu & Li, 2020). However, these theories could not explain everything we observed in experiments for ranking distillation, especially, why the listwise distillation is necessary in Section 5.4. Here we provide a new theory that helps to explain BAR effectiveness.

**Theorem 1.** *There exists a way to combine the teacher prediction score and the label to train a student model, such that when the teacher and the student have exactly equivalent capability, the student model can outperform or perform as well as the teacher model after being trained with the same amount of resource.*

It's easy to show the necessity: we can just train the student model with the true label with $\alpha = 0$ in Eq.(4) and the same computation resource and it then performs as well as the teacher model. In the Appendix Section A.1, we show the sufficiency of the theorem when there are data points that are hard to fit, or that have erroneous labels.

Here we give an illustrative example of Theorem 1. Consider a training set of three data points with a single input feature $x = -1, 0, 1$ and corresponding labels are $y = 2, 1, 2$, shown as red points in Fig. 4 and a test set of two data points $(0, 0)$ and $(0.5, 1)$ in cyan. We fit the training points with a single-parameter nonlinear model: $f(x, b) = 2|x - b|$, shown in a green dashed line in Fig. 4. To this model, two data points $(-1, 2)$ and $(1, 2)$ are easy-to-fit and one data point $(0, 1)$ is hard or mistakenly labeled as it is inconsistent with $(0, 0)$ in test set.

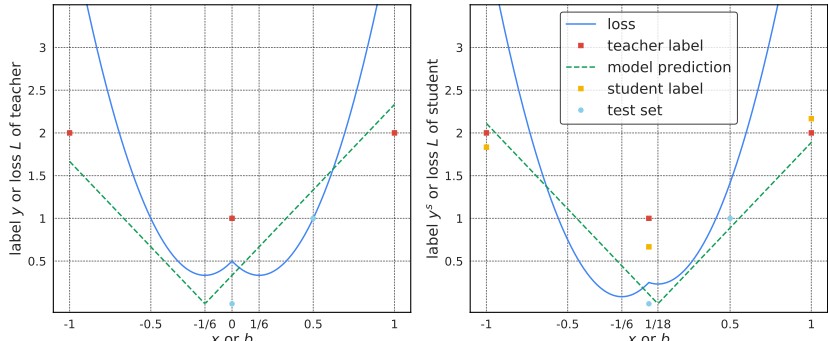

Figure 4: An illustrative example of born again distillation. Given the training data set $(x, y)$ (red squares), loss functions $\mathcal{L}(b)$ (blue solid lines) and optimized models $f(x)$ (green dashed lines) are trained with the original label $y$ in red for the teacher model on *Left*, and are trained with the combined label $y^s = (y + f^t(x))/2$ in yellow for the student model on *Right*. The models are then evaluated with the test data set in cyan dots.

By minimizing a mean squared error loss, $\mathcal{L}(b) = \frac{1}{2} \sum_{i=1}^{3} (y_i - f(x_i, b))^2$, shown as the blue line in Fig. 4, we find two nontrivial solutions at $b = -1/6$ and $1/6$. Suppose we obtain a teacher model with $b = -1/6$ and scores $f^t = 5/3, 1/3, 7/3$ for the three inputs and then train a student model following the BAR method with $\alpha = 0.5$, which is equivalent to having the student labels $y^s = (y + f^t)/2 = 11/6, 2/3, 13/6$ respectively, shown as the yellow points in Fig. 4. There are again two nontrivial solutions with $b = -1/6$ and $1/18$ for the student. Ignoring the trivial solution at $b = -1/6$, we find the student solution at $b = 1/18$ achieves a better performance by having a smaller mean square error metric $MSE = 1/81$ on the test points than $MSE = 1/9$ of the teacher solution at $b = -1/6$.

**Applying Theorem 1 to learning to rank.** Some of the labels in the LTR dataset could be error-prone as a result of the pointwise evaluation nature of human rating. In the listwise ranking sense: (1) Some of the labels may be error-prone, such that a document labeled with 2 might not be more relevant than a document labeled with 1; (2) The graded relevance labels are discrete and may not be able to fully capture the relevance in-between like 1.5; (3) A listwise loss only needs the comparison within each individual list and this can alleviate the impact of noisy labels comparing with a pointwise loss. As a result, the listwise distillation is able to capture these mistakenly labeled data points and thus performs significantly better than the examples using pointwise distillation. This deduction is also consistent with the findings in Section 5.4: the most effective distillation is on models trained with the listwise Softmax loss.

**The regularization view.** Even if the human labels are not noisy but just contain some hard examples, Theorem 1 implies that combination like Eq.(4) reduces overfitting on hard examples and plays a regularization role. Qin *et al.* (Qin et al., 2021) show that one major bottleneck of modern neural rankers is overfitting, due to the lack of very large-scale ranking datasets and effective regularization techniques. By examining Eq. 4, the second term can be treated as regularization to the original objective. We empirically show that the BAR framework works as a very effective regularization technique for LTR problems. As seen in Figure 3 (Right), the BAR student gets lower training data ranking metrics and higher validation data ranking metrics during training, the desired behavior of effective regularization.

## 7 CONCLUSION

We propose Born Again neural Rankers (BAR) and further push the limits of neural ranker effectiveness without sacrificing efficiency. We show that the key success factors of BAR lie in a proper teacher score transform and a listwise distillation approach specifically designed for ranking problems, which do not follow the common assumptions made in most knowledge distillation work. We further conduct theoretical analysis on why BAR works, filling the gap between existing knowledge distillation theories and the learning to rank setting. As promising directions for future work, we consider advanced teacher score transformations for BAR, and label noise reduction techniques.

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

# A APPENDIX

## A.1 PROOF OF THEOREM 1

In this section, we show the student model trained with Eq.(4) can outperform the teacher model with the same capacity at an optimal $\alpha^*$ if there exist hard examples or the labels are noisy.

We first formulate the problem and give the mathematical definitions of "hard examples" and "noisy labels". When training converges, a deep neural network teacher model $f(\cdot, \theta^t)$ approximately satisfies

$$\frac{\partial \mathcal{L}}{\partial \theta_\gamma} = \frac{1}{|\Psi|} \sum_{(\mathbf{x},\mathbf{y}) \in \Psi} \sum_{i=1}^{n} \frac{\partial l(\mathbf{y}, \mathbf{s})}{\partial s_i} \frac{\partial f(\mathbf{x}, \theta)_i}{\partial \theta_\gamma} = 0 \tag{11}$$

for any parameter $\theta_\gamma$ in the total $P = |\theta|$ trainable parameters. The first derivative on the right hand side is the loss function dependent gradient. For example:

- Mean Squared Error loss: $l(\mathbf{y}, \mathbf{s}) = \frac{1}{2} \sum_{i=1}^{n} (y_i - s_i)^2$, we have

$$\frac{\partial l(\mathbf{y}, \mathbf{s})}{\partial s_i} = -(y_i - s_i);$$

- Pointwise logistic loss: $l(\mathbf{y}, \mathbf{s}) = -\sum_{i=1}^{n} y_i \ln p_i + (1 - y_i) \ln(1 - p_i)$ with $p_i = \frac{\exp(s_i)}{1 + \exp(s_i)}$, we have

$$\frac{\partial l(\mathbf{y}, \mathbf{s})}{\partial s_i} = -(y_i - p_i);$$

- Softmax loss in Eq.(5), we have

$$\frac{\partial l(\mathbf{y}, \mathbf{s})}{\partial s_i} = -(y_i - p_i \sum_{k=1}^{n} y_k).$$

If we normalize the labels by having $\sum_i y_i = 1$ in Softmax loss and define the model gradients on parameter $\theta_\gamma$ at data point $i$, $G_{\gamma,i} = \partial f(\mathbf{x}, \theta)_i / \partial \theta_\gamma$ as a $P \times n|\Psi|$ matrix, we then have ranking model solutions satisfying $P$ equilibrium equations,

$$G \cdot (\mathbf{y} - \mathbf{p}) = 0, \tag{12}$$

with $\mathbf{p}$ as a vector of model predictions depending on different losses as shown in above examples. When the rank of $G$ is $n|\Psi|$, we only have trivial solutions with $\mathbf{p} = \mathbf{y}$. However, noting that the matrix $G$ is a nonlinear function of $\theta$, in general we have the rank of $G$ is $P < n|\Psi|$ with non-trivial solutions $\mathbf{p} \neq \mathbf{y}$ living in the null space of $G$.

To such a solution, we can distinct hard and easy problems in the training dataset by quantifying the deviations from the target labels in the trained teacher model $\|y_i - p_i\|$: the larger this deviation is, the harder the corresponding data point is. These data points in the training set could be hard in two different senses:

- Case *One* – Hard examples: they are hard in nature, meaning that the trained model does not capture the most deterministic feature of the data points.
- Case *Two* – Noisy labels: they could be mistakenly labeled, meaning that there exists a ground-truth model and the ground-truth model predicts results different from the labels for these data points.

When we train a student model from scratch with combined teacher predictions and labels following Eq.(4) with the teacher model solution $\mathbf{p}^t$ as $g(f(\mathbf{x}, \theta^t))$ so that $G \cdot (\mathbf{y} - \mathbf{p}^t) = 0$, the student solution should then also satisfy Eq.(12) but with a new set of labels $\mathbf{y}^s = (1 - \alpha)\mathbf{y} + \alpha \mathbf{p}^t$. First of all, it's easy to show that the teacher model with $\mathbf{p}^t$ is a solution. But for a complex nonlinear model trained from scratch, it would be almost impossible to end up with the same solution as the teacher in general, especially when the labels are changed, which determine the initial gradients. We thus focus on the general solutions different from $\mathbf{p}^t$.

In Case *One*, the model output will be insensitive to the change of parameters for the hard points at initial training stage. After easy points are fit well, $y - p^t \approx 0$, the training enters a stage of fitting the hard points with irrelevant features, which directly leads to overfitting. So in practice, we use predictions $p^t$ from a teacher regularized with early-stopping as the distillation labels for students. With such a teacher, the initial gradients of the student model are still dominated by easy data points $p_i \approx p_i^t$ and in the later stage the contributions of the hard data points $y_i^s - p_i = (1 - \alpha)(y_i - p_i^t) + (p_i^t - p_i) \approx (1 - \alpha)(y_i - p_i^t)$ are lowered by fraction $1 - \alpha$ compared to the training of the teacher model. The overfitting effect will thus be weakened for the student model. In this case, the student model can achieve a better performance on the test data than the teacher model with $\alpha > 0$.

In Case *Two*, the final solution of the teacher model will be balanced by contributions of the correct labels and wrong labels: let's say there are $m$ out of $n$ training data points mistakenly labeled and most are correct labels $m \ll n$: on average, the teacher model prediction will deviate from the label by $(n - m)|y^c - p^t| \sim m|y^w - p^t|$, where $y^c$ are correct labels and $y^w$ are wrong labels. We thus have the teacher predictions deviating from the correct labels by $|p^t - y^c| \sim \frac{m}{n} y^c$ and from the wrong labels by $|p^t - y^w| \sim -(\frac{m}{n} - 1)y^w$, so the student labels are $y^s = y - \alpha \frac{m}{n} y$ for correct labels and $y^s = (1 - \alpha)y + \alpha \frac{m}{n} y$ for wrong labels. After convergence, the student predictions will thus be $p^s \approx y + (1 - \alpha)\frac{m}{n} y - \alpha \frac{m^2}{n^2} y$ for correct labels and $p^s \approx (1 - \alpha)\frac{m}{n} y - \alpha \frac{m^2}{n^2} y$ for wrong labels. When $\alpha^* = \frac{n}{n+m}$, student predictions on examples with correct labels are consistent with labels and on examples with wrong labels are approximately independent of the label. Assuming that there is no correlation of making the wrong labels in training and test sets, then the student models validated on the test set will result in better metrics than their teachers as they are making predictions closer to the correctly labeled data points.

