# OpenReview forum: "Born Again Neural Rankers"
_ICLR.cc/2022/Conference — ICLR 2022 Submitted_

### Official Review · Reviewer_wTyv · 2021-10-20

**Correctness:** 4
**Technical Novelty And Significance:** 3
**Empirical Novelty And Significance:** 3
**Recommendation:** 8
**Confidence:** 4

**Main Review:**

The paper introduces Born Again neural Rankers (BAR) for Learning to rank (LTR). Unlike traditional Knowledge Distillation (KD) which usually seeks a trade-off before performance and efficiency, BAR targets for better performance without increasing the model capacity.

In particular:

1. The authors clearly state two important design choices: 1) a listwise Softmax loss function and 2) a teacher score transformation. In additional, they also explore other alternatives and empirically show why the current choice is warranted.

2. Extensive experiment studies are conducted to demonstrate the effectiveness of the proposed method.

3. The authors also propose an explanation to better understand the superior performance of BAR.

There are several places where I am not certain, or maybe can make the paper better:

1. In the experiment there are several ensembles considered, including DASALC-ens and BAR-ens. For a more fair comparison, I wonder whether the authors can also compare with ensembles of \lambdaMART, since it achieves two best performances across 9 settings.

2. In (4) there is a hyperparameter \alpha, and in (7) there is a and b for the affine transformation. The experiment results shown in Table 1 only uses fixed values for these hyperparameters. It might be worth to see if we tune more on these three hyperparameters, how much more gain the model can achieve.

3. Is such complexity necessary or is more complexity helpful? I do not expect the authors to address this point in this paper, but it would be helpful to share some thoughts if they have. Currently, we have the student model the same complexity as the teacher. However, is such complexity really necessary? Can we have a less complexity student while also achieve superior performance? From another perspective, if we have a more complexity student, with the help of knowledge distillation, is it able to perform even better?

**Summary Of The Paper:**

The paper introduces Knowledge Distillation (KD) to Learning to rank (LTR), adapting the idea of Born Again Networks (BAN). The propose model, Born Again neural Rankers (BAR), combines an appropriate teacher score transformation and a novel listwise distillation framework. The authors conduct thorough experiment studies to demonstrate the superior performance of the BAR, and also provide explanations in terms of why it works.

**Summary Of The Review:**

I believe this paper is well written with clear statement of their research questions, extensive experiments, and rigorous arguments. I suggest to accept this paper to ICLR.

---

> ### Author Response · Authors · 2021-11-15
> **Response to Reviewer wTyv**
>
> Thank you very much for your review.
>
> Ensembling lambdaMART models were done in (https://openreview.net/forum?id=Ut1vF_q_vC) Appendix B.6, where there was limited benefit. The hypothesis was “model ensembles tend to be more effective for neural rankers with stronger stochastic nature”, which we concur.
>
> In terms of the hyperparameters - we agree that further tuning the hyperparameters might produce better results. We found it empirically appealing that state-of-the-art results can be achieved without heavy tuning, but we agree that it would be interesting to further tune them, ideally in a more efficient way, e.g., using AutoML.
>
> In this paper we focus on the born again setting. If the model is smaller, it’s knowledge distillation and will generally perform worse than the teacher model. The major goal of this paper is to push state-of-the-art results on learning to rank benchmarks, using the neural methods in particular. It would be interesting to focus on effectiveness-efficiency trade-offs. In terms of using a larger model - it's an interesting direction. Larger models tend to be more data hungry, which might propose some interesting research questions.

---

### Official Review · Reviewer_TTzh · 2021-11-02

**Correctness:** 3
**Technical Novelty And Significance:** 2
**Empirical Novelty And Significance:** 2
**Recommendation:** 3
**Confidence:** 4

**Main Review:**

Strengths:
The paper investigates an interesting and under explored area of model distillation and learning to rank, and is well written and easy to follow. Experiments are done on large real world datasets and compare against leading baselines from both gradient boosting and deep learning domains.

Weaknesses:
My major concern is with the novelty. The proposed approach is a relatively straightforward application of distillation with off-the-shelf loss and optimization. I also have a concerns about section 4.3, ReLU is an odd transformation to use in the learning to rank setting as it squashes all negative scores to 0 making that part of the ranking indistinguishable. This can significantly affect teachers accuracy and also introduces many ties into the students loss. The justification in 4.3 does not support such a transformation, and in section 5.1 authors actually use identity for Web30K and Yahoo datasets, and divide scores by 100 for Istella. These choices again seem hand picked and non-principled, and further justification is needed here since authors claim that normalising teachers scores is important for distillation to work.

**Summary Of The Paper:**

The paper proposes a distillation approach for the learning to rank setting. Specifically, the "born again neural ranker" approach is investigated where both teacher and student models are identically parameterised neural networks. Authors propose a listwise loss to incorporate scores from the teacher model during student optimization and show that this leads to improvements on three real world datasets.

**Summary Of The Review:**

The paper explores an important area in learning to rank but the proposed approach lacks novelty and makes multiple ad hoc choices that need further justification.

---

> ### Author Response · Authors · 2021-11-15
> **Response to Reviewer TTzh**
>
> Thank you very much for your review. We want to clarify a misunderstanding in terms of ReLU. As stated in the paper: “unlike classification problems where predictions are non-negative probabilities, teacher ranker scores can be negative, which will make many ranking losses ill-behaved”. So the use of ReLU is to purely avoid making many loss functions invalid, such as cross-entropy based losses. To avoid making “ranking indistinguishable” is exactly why we introduced hyperparameter b in Eq7. Eq7 is a general proposal for ranking distillation, where (we believe) we considered most common potential issues of the teacher ranker, since unlike classification, there are no constraints on teacher score.
>
> On the other hand, we have used Softmax transformation instead of ReLU to avoid teacher labels being negative. Our experimental results show that the Softmax transformation of teacher labels is not effective, though it does not have the ranking indistinguishable problem for negative labels. This is an interesting observation that we saw in other applications too and worth exploring more in the future.
>
> Parameters including a and b are treated as hyperparameters in this paper. How they are set depends on the teacher ranker behavior and needs to be tuned on the validation set. In fact, we showed that by chance, we did not need much tuning for two teacher rankers to achieve state-of-the-art results. However, these are purely empirical, and due to the lack of theoretical guarantee, section 4.3 focuses on a general statement. In fact, the teacher ranker on Istella dataset has a wider value range and negative scores. Dividing score by 100 resulted from hyperparameter tuning.
>
>
> In terms of novelty, pushing state-of-the-art results on benchmark learning to rank datasets is a difficult task. We draw ideas from the vast literature of knowledge distillation and make concrete adjustments for ranking problems with solid improvements over the previous state-of-the-art, which we believe has solid contribution to the literature.

---

### Official Review · Reviewer_7kAC · 2021-11-03

**Correctness:** 2
**Technical Novelty And Significance:** 2
**Empirical Novelty And Significance:** 2
**Recommendation:** 3
**Confidence:** 4

**Main Review:**

**Strengths**

* The paper clearly motivates the need for distillation specific to LTR problems, explained the need for a score transformation layer and recommends one way (BAR) to achieve good transfer learning on 3 public datasets.
* The primary contribution, i.e. transforming teacher scores using a ReLU + Affine layer and using listwise loss for the student model are simple, easy to understand and replicate. The effect of hyperparameters could have been studied better.

**Weaknesses**

* The paper is very light on experimentation. e.g. It would have been good to see the comparison of listwise vs pointwise teachers on more than one dataset - ideally with 3-5 neural architectures that have been used in the literature. What was the effect of the scaling factor in the Istella dataset?  How sensitive is $\alpha$ across datasets? Does early stopping play a role in the generalization performance? Does $L_2$ affect the regularization ability of BAR?
* Lack of theoretical justification of the BAR choices. The authors claim "theoretical explanations" as a contribution of the paper. Yet, question like why ReLU, why affine transformation, how to fit parameters of the transformation on new datasets are not justified by theory. The section on Understanding BAR merely provides an intuition and one hypothesis explaining the trend.


**Summary Of The Paper:**

Knowledge distillation (KD) is a popular technique to trade-off between performance and efficiency. Student models trained with KD usually work better than those trained from the original labels. However, applying KD to ranking problems is not well studied. One earlier work (Tang & Wang, 2018) on ranking distillation proposes pointwise logistic loss for distillation after unsuccessfully attempting pair-wise approaches on the teacher's top K ranking. The authors take on the challenge of dealing with the following characteristics of the ranking models:
* The ranking scores are neither calibrated nor normalized. They do not directly convey information about the underlying distribution.
* They care about the relative order of score and do not make an iid assumption over documents.
* Large teacher models tend to overfit because it is difficult to apply techniques like data augmentation & small datasets.

The authors extend BAN in which student models are configured with the same capacity as teachers and propose BAR. BAR is referred to the specific setting where ranking loss with original labels, listwise distillation on teacher score, and tunable affine teacher score transformation. Unlike typiical KD models, the primary goal of the paper is to improve ranking performance over state-of-the-art teacher rankers.

**Summary Of The Review:**

The authors empirically show improvements on 3 public LTR dataset using the BAR method over a neural baseline. To clearly support their research questions, they should continue experimenting with a few more baseline neural models to understand the effectiveness of BAR. Additionally, the ablations and design choices for transformation & loss functions should be done on more than one dataset to be able to conclusively make the claims made in the paper. The biggest gap is however, the lack of theoretical justification on why the specific choices in BAR are needed. The paper can just have empirical evidence to show one transformation that works well across datasets. However claiming that BAR is a novel listwise distillation framework with theoretical justification is far-fetched.

---

> ### Author Response · Authors · 2021-11-15
> **Response to Reviewer 7kAC**
>
> We strongly disagree with comments.
>
> For the comment “the paper is very light on experimentation”, we disagree. Our methods are validated on 3 largest LTR datasets. The review picks on the hyperparameters such as \alpha where we only showed results on 1 dataset. In reality, hyperparameters need to be tuned in general. The review asks for an exhaustive study on 3-5 architecture x 3 datasets, which looks demanding for a paper. The other way to question this is whether the current experimental results are adequate to support the paper. We think they are adequate.
>
> As explained in the paper, ReLU is used since many popular loss functions will be invalidated with negative (teacher) labels. Parameters of the affine transformation are treated as hyperparameters and there are just two of them. In fact we showed that we did not need much tuning to achieve state-of-the-art results, which we believe is a benefit, not a drawback.
>
> For the comment “how to fit parameters of the transformation on new datasets”: The widely adopted approach is to choose these hyperparameters based on a validation data set.

---

### Decision · Program_Chairs · 2022-01-20

**Decision:**

Reject

**Comment:**

The authors are strongly encouraged to elaborate further about the novelty of their method, as well as to give detailed (either theoretical or experimental) justifications for the design choices they make within the paper. Finally, the paper could benefit from additional experiments, as outlined in the reviews.